# Are Women with Polycystic Ovary Syndrome at Increased Risk of Alzheimer Disease? Lessons from Insulin Resistance, Tryptophan and Gonadotropin Disturbances and Their Link with Amyloid-Beta Aggregation

**DOI:** 10.3390/biom14080918

**Published:** 2024-07-28

**Authors:** Joachim Sobczuk, Katarzyna Paczkowska, Szymon Andrusiów, Marek Bolanowski, Jacek Daroszewski

**Affiliations:** 1Department of Endocrinology, Diabetes and Isotope Therapy, University Clinical Hospital, 50-367 Wroclaw, Poland; 2Endocrinological Ambulatory Care, 62-700 Turek, Poland; 3Department of Neurology, Wroclaw Medical University, 50-556 Wroclaw, Poland; 4Department of Endocrinology, Diabetes and Isotope Therapy, Wroclaw Medical University, 50-556 Wroclaw, Poland

**Keywords:** amyloid, tau protein, alzheimer disease, polycystic ovary syndrome, neurodegeneration, kynurenine pathway, serotonin, tryptophan, luteinizing hormone, insulin resistance, notch, stAR

## Abstract

Alzheimer disease, the leading cause of dementia, and polycystic ovary syndrome, one of the most prevalent female endocrine disorders, appear to be unrelated conditions. However, studies show that both disease entities have common risk factors, and the amount of certain protein marker of neurodegeneration is increased in PCOS. Reports on the pathomechanism of both diseases point to the possibility of common denominators linking them. Dysregulation of the kynurenine pathway, insulin resistance, and impairment of the hypothalamic-pituitary-gonadal axis, which are correlated with amyloid-beta aggregation are these common areas. This article discusses the relationship between Alzheimer disease and polycystic ovary syndrome, with a particular focus on the role of disorders of tryptophan metabolism in both conditions. Based on a review of the available literature, we concluded that systemic changes occurring in PCOS influence the increased risk of neurodegeneration.

## 1. Introduction

Alzheimer disease (AD) was first described in 1906 [1], and it is the most common cause of dementia, responsible for approximately 50–70% of cases [2]. The prevalence of the disease increases with age, and is estimated to be between 10.3% and 19.5% for men and women at age 45, respectively, and 11.6% and 21.1% at age 65 [3]. Currently, the main role in the pathogenesis of Alzheimer disease is attributed to tau protein and amyloid-beta [4]. The causal therapy has not yet been established however, monoclonal antibodies targeted specifically against beta-amyloid were introduced in recent years as a potential treatment, but the results of the therapy were not as satisfactory as they had been expected [5,6]. The same is true for polycystic ovary syndrome (PCOS), which was first described by Stein and Leventhal in 1935 and occurs in 6–20% of reproductive-age women [7,8,9]. The pathomechanism is still not fully understood, and, as a consequence, causal treatment is not available.

Both disease entities, seemingly unrelated, share common features, such as chronic inflammation, mitochondrial dysfunction, and insulin resistance, exacerbating the pathophysiological processes. Moreover, cardiovascular disease, hypertension, diabetes mellitus type 2, or obesity, which are much more common in PCOS, are risk factors for AD as well.

In recent years, a number of studies have postulated the role of disruption of various metabolic pathways in the pathogenesis of both diseases; it turns out that both entities share common pathophysiological disturbances. Taking that into account, it might be supposed that processes occurring in PCOS women may promote the onset of neurodegeneration in later life. This hypothesis is supported by the fact that amyloid-beta is an immunoactive peptide, and its levels are increased in chronic inflammation. As a component of the innate immune system, its levels rise in response to chronic inflammation [10]. In addition, higher concentrations of amyloid precursor protein (APP) were found in the plasma of PCOS patients [11].

It is worth noting that most studies regarding PCOS are conducted in premenopausal patients, but on the other hand, the process of neurodegeneration in AD, excluding early-onset form, usually develops fully in older age. To the best of the authors’ knowledge, it has not been studied whether the diagnosis of PCOS at a young age implicates an increased risk of Alzheimer disease in later life.

In this review, the authors analyze available literature for possible mechanisms linking the two conditions, suggesting that patients with PCOS might be at increased risk of AD.

## 2. Outline of the Pathophysiology of Alzheimer Disease

Due to the age of onset of the disease, two forms are distinguished: early-onset, occurring before the age of 65, and late-onset, occurring after the age of 65. These forms significantly differ in terms of prognosis, symptomatology, and pathogenesis [4]. During the course of the disease, there is a progressive atrophy of specific areas of the brain, beginning with the intraparietal cortex, the hippocampus, the amygdala, and then the association cortex, primarily within the frontal, parietal and temporal lobes [12]. Although the exact patomechanism of AD is not well understood, there are many hypotheses describing underlying processes. It seems that the initiating process of the neurodegeneration cascade is the deposition of amyloid-beta in senile plaques. The amyloid precursor protein (APP) is cleaved via the enzyme beta-secretase to an 89 or 99 amino acid peptide, which is then cleaved by gamma-secretase (BACE1) to amyloid-beta 1–40 (Abeta40) or amyloid-beta 1–42 (Abeta42). The former is well soluble, but the latter easily form deposit. The variant polymorphism of APP, the catalytic domain of gamma-secretase-presenilin 1 and 2, and the presence of the APOE e4 allele determine which version of amyloid-beta will be formed in greater amounts [13]. Abeta15, Abeta16, Abeta17 and Abeta38 have also been described in the brains of AD patients [14]. The process of forming senile plaques is the most characteristic hallmark of AD [15]. However, the amount of deposit does not correlate with the degree of cognitive deficits of patients [16]. Amyloid-beta also forms soluble oligomers, which most likely initiate the pathology of tau protein in an unclear mechanism and, as a result, cause neuronal degeneration [17,18]. Oligomers contribute to neuroinflammation and synapsotoxicity [19], and their concentration correlates with cognitive deficits [20].

Tau protein is an essential protein for the proper process of microtubule formation; it participates in the process of axon elongation and in the formation of new intercellular connections within dendrites. In AD, there is excessive phosphorylation of this protein, and it begins to form neurofibrillary tangles (NFTs), which, when deposited in the neurocyte, lead to neuronal atrophy. This process takes place in the brain areas mentioned above [13,21], and the amount of NFT deposits located in the brain correlates with the degree of cognitive deficit [22].

Impairment of the cholinergic system, which is particularly involved in cognitive processes, especially memory, also plays a significant role in the Alzheimer disease pathogenesis [23]. In the course of AD, the basal nucleus of Meynert (NBM) located in the basal part of the forebrain is primarily damaged [24]. NBM projects to numerous areas of the cerebral cortex, hippocampus cortex, and intraparietal cortex. Progressive atrophy and axon damage is observed in AD in these areas; the process is correlated with amyloid-beta and tau protein burden [25,26,27].

It is suspected that the occurrence of Alzheimer disease requires multifactorial dysfunction along with the presence of appropriate environmental conditions. Numerous genes involved in the pathogenesis of Alzheimer disease have been described so far, including, but not limited to, mutations in amyloid precursor protein (APP), presenilin 1 and 2 (PSEN1 and PSEN2), and carrying the E4 allele of Apolipoprotein (ApoE) genes. Mutations in the ApoE gene are considered the most significant risk factor for the disease, increasing the risk of its development by a factor of 3–4. Finding mutations in these genes is primarily applicable in early-onset Alzheimer disease [28,29,30,31]. The role of genetic factors acting protectively on the central nervous system is also being considered to be involved in AD pathogenesis [32,33].

In recent years, the role of neuroinflammation has been increasingly emphasized as a significant factor causing neuron damage in the course of Alzheimer disease; microglia activation, in particular, can lead to either neuroprotective or neurodegenerative effects [34,35,36].

Despite decades of research and a huge amount of data, there is still debate about the causality and exact role of amyloid-beta and tau protein in the pathogenesis of the disease. This is reflected in widespread research regarding biochemical and imaging biomarkers use in diagnostics, whose presence, confirming the pathology of beta-amyloid and tau protein, strongly facilitates the diagnosis of Alzheimer disease [37].

## 3. Outline of the Pathophysiology of PCOS

PCOS is a condition in which hyperandrogenemia, menstrual irregularity, and polycystic ovary morphology occur. While the exact pathogenesis is still unknown, there is considerable evidence linking the development of PCOS to insulin resistance (IR) [38]. It is present in 50% to 70% of cases [39] and appears to be immanent to PCOS, as it occurs regardless of BMI [40,41]. Patients with PCOS, in addition to the above-mentioned traits may exhibit a variety of metabolic, dermatological, and psychiatric health problems [42]. Different phenotypes depending on the presence of particular traits, and many diagnostic criteria were developed as a result of heterogeneity in symptomatology [8]. The diagnosis of PCOS is usually based on the Rotterdam criteria, which requires the presence of two out of three factors: hyperandrogenism, ovulatory dysfunction, and polycystic morphology of the ovaries on ultrasound examination [43].

Apart from the above key inherent features of PCOS, numerous phenomena modifying its course have been studied. Patients experience reprogramming of the pulse generator in the hypothalamus, which leads to abnormal secretion of gonadotropins with increased secretion of luteinizing hormone (LH) [44]. This promotes excessive ovarian androgen production, which disrupts ovarian follicle maturation and, as a consequence leads to ovulation disorders and characteristic polycystic ovarian morphology. Moreover, IR leads to compensatory hyperinsulinemia and additional stimulation of the LH receptor by insulin; this process also leads to androgen overproduction by theca cells in the ovary [45]. However, PCOS is not just a gynecological problem. Although PCOS is typically diagnosed during reproductive age, it impacts patients’ health throughout their entire lifetime. Current diagnostic criteria do not take into account metabolic changes, which are an integral component of the syndrome.

In PCOS, mitochondrial dysfunction and chronic inflammation are observed [46]. Proinflammatory cytokines (TNF-α, CRP, IL-6, IL-8, and IL-18) level elevation is also observed [46]. Concomitant microflora dysbiosis is postulated to promote chronic low- grade inflammation and IR [47,48]. This might explain why the population of PCOS patients has a higher risk of metabolic syndrome and its complications, such as diabetes mellitus type 2, hypertension, dyslipidemia, non-alcoholic fatty liver disease (NAFLD), obesity, cardiovascular disease, and endothelial dysfunction [49,50,51,52]. Moreover, there is also an increased incidence of depression, eating disorders, and sleep disorders [53,54,55] (Figure 1).

This indicates the presence of significant metabolic abnormalities underlying the condition, manifested by dysfunctions of many systems and organs, including the central nervous system. The role of unhealthy lifestyles and chemical endocrine disruptors is also postulated [56,57].

Diabetes Mellitus

Advances in the analysis of metabolic markers make it possible to study amino acid profiles. Elevated levels of branched-chain amino acids (BCAA) are correlated with the presence of metabolic abnormalities in PCOS and may serve as an early biomarker of the disease [58,59]. The claim that systemic metabolic rather than exclusively gynecological disorders underlie PCOS is consistent with the fact that a male equivalent of PCOS has been postulated. Male relatives of PCOS women more often present features of the metabolic syndrome, insulin resistance, early-onset androgenic alopecia, increased risk of cardiovascular disease, and benign prostatic hyperplasia when compared to the general population [60,61,62].

New diagnostic and laboratory techniques, as well as research related to understanding the processes underlying PCOS and the metabolic syndrome, are contributing to the discovery of new diagnostic markers, such as changes in amino acid metabolism or organokines [63,64,65,66,67].

## 4. Physiology of Tryptophan Metabolism

Tryptophan (Trp) is classified as an essential amino acid, which can be only obtained by the body through external supply. Sources of Trp include poultry, beef, pork, lamb, nuts, legumes, and dairy products [68]. The daily requirement for this amino acid is 4 mg/kg body weight per day [69]. It belongs, along with tyrosine and phenylalanine, to the aromatic amino acid (AAA) group [70]. Trp is used as a protein building block and is a substrate for numerous signaling substances; additionally, its metabolites play a regulatory role in various body processes. Only 1% of Trp supply absorbed in the gastrointestinal tract is used as a protein building block; the remainder is metabolized via other pathways [71,72,73]. Thus, ingested Trp is used to build proteins, incorporate them into the kynurenine pathway, and produce serotonin and melatonin.

### 4.1. Production of Serotonin and Biogenic Amines

Approximately 5% of absorbed tryptophan is converted to serotonin [74], and this process occurs mainly in the enterochromaffin cells of the intestine. Through hydroxylation, Trp is firstly converted to 5-hydroxytryptophan and then to 5-hydroxytryptamine (serotonin) by the enzyme aromatic amino acid decarboxylase [75]. Overall, 95% of the serotonin produced in the body is secreted in the gut, with only 5% synthesized locally in the central nervous system (CNS). Serotonin plays a key role as a neurotransmitter in the enteric nervous system, regulating gastrointestinal motility. Serotonin cannot cross the blood-brain barrier (BBB), so it must be synthesized within the CNS from Trp. Trp is the only amino acid in the bloodstream that is transported in an albumin-bound form, allowing it to cross the BBB via the LAT1 transporter [76,77].

### 4.2. Indole Derivatives

Part of the ingested tryptophan is processed by the commensal gut microbiota to produce its own proteins, polyaromatic hydrocarbons, indole derivatives, and a small amount of tryptamine. These compounds regulate intestinal permeability, immunity, inflammation, and insulin sensitivity [78,79,80,81].

### 4.3. Kynurenine Pathway

About 95% of absorbed tryptophan is metabolized via the kynurenine pathway [82] (Figure 2); the initiating and rate-limiting step in this pathway is the conversion of tryptophan to kynurenine by tryptophan 2,3-dioxygenase (TDO) and indoleamine 2,3-dioxygenase (IDO) types 1 and 2. TDO is primarily found in the liver and is responsible for producing about 90% of kynurenine, while IDO is found in immune cells, adipose tissue, kidneys, the brain, and pancreatic beta cells [83]. The activity of TDO and IDO, and thus the extent of Trp incorporation into the kynurenine pathway, is reflected by the Trp/kynurenine ratio [84].

Kynurenine is metabolized in several directions. It is converted to kynurenic acid (KYNA) by kynurenine aminotransferase (KAT). On the other hand, with the involvement of kynureninase (KYNU), it is converted to anthranilic acid (AA), which undergoes conversion to 3-hydroxyanthranilic acid (3-HAA) through nonspecific hydroxylation. Via kynurenine 3-monooxygenase (KMO), kynurenine is converted to 3-hydroxykynurenine (3-HK). At this stage, the pathway diverges again: 3-HK can be a precursor for xanthurenic acid (XA) (a reaction catalyzed by KAT) or for 3-HAA (a reaction catalyzed by KYNU), which is then converted to picolinic acid or quinolinic acid (QA). QA is converted via quinolinate phosphoribosyltransferase (QPRT) to NAD(P)H in mitochondria [85,86,87,88].

The various metabolites and enzymes of the kynurenine pathway have diverse biological activities, and their roles in specific diseases are currently under investigation. QA has neurotoxic effects on CNS [89], but, on the other hand, kynurenic acid has neuroprotective and cardioprotective effects [90]. Moreover, the role of KYNA in carcinogenesis has been postulated; however, its concentrations vary in different types of cancer, making this role unclear [89,91,92]. Anthranilic acid contributes to the development of depression through inflammatory processes [93]. 3-hydroxykynurenine (3-HK) also exhibits neurotoxic properties [94]. Xanthurenic acid (XA) concentrations decrease in response to improvement in glucose tolerance after bariatric surgery; in animal models, administration of XA results in a decrease in arterial pressure [95,96]. Picolinic acid has antiproliferative properties, suggesting an anticancer effect [96,97].

Inflammation is connected to an increase in the activity of individual enzymes in the kynurenine pathway and a change in the concentrations of its various metabolites. Enhanced activation of the discussed pathway is observed in obesity, chronic obstructive pulmonary disease [98], diabetes mellitus type 1 and type 2 [87,99], atherosclerosis accompanying chronic kidney disease [100], cardiovascular diseases [101,102], neoplasms [103], and neurodegenerative diseases [104]. Therefore, it is postulated that inhibitors of specific enzymes involved in this pathway might be used as therapeutic agents for these diseases. The potential of IDO1 inhibitors for anticancer treatment is being explored, leveraging the fact that inhibiting this enzyme reduces immunotolerance induced by Trp metabolites [105]. Kynurenine monooxygenase (KMO) inhibitors have been utilized in the treatment of acute pancreatitis in animal models [106]. Their potential role in the treatment of neuropathic headaches has been suggested, attributed to their ability to decrease the neurotoxic quinolinic acid [107]. Furthermore, the potential of KMO inhibitors in the treatment of rheumatologic diseases is also under consideration [108].

## 5. Common Denominators of PCOS and Alzheimer Disease

### 5.1. Insulin Resistance and Metabolic Disorders

There are many risk factors for Alzheimer disease, including smoking, depression, low levels of education, hearing loss, and other conditions that weaken social ties, as well as head injuries and pollution [2,109]. Risk is higher also in metabolic diseases, such as type 2 diabetes, insulin resistance, dyslipidemia, hypertension, and hyperhomocysteinemia [110,111,112,113]. Obesity also promotes the onset of neurodegeneration in middle-aged people; however, later in life, the correlation between AD incidence and obesity is not present [114].

Insulin resistance (IR) is a significant metabolic disorder in PCOS pathogenesis present in the vast majority of subjects that also serves as a risk factor for AD [39].

PET is a valuable source of data indicating involvement of particular structures of the central nervous system in AD pathology. Much recent research uses Pittsburgh compound B (PiB-PET), a radiotracer accumulating in senile plaques. It was found that the level of insulin resistance expressed by the HOMA index correlated positively with the amount of amyloid-beta deposition assessed using PiB-PET in AD-related areas in normoglycemic middle-aged subjects [115] (Table 1). An observational study of a Finnish population found that individuals with insulin resistance in middle age were more likely to have amyloid-beta deposits imaged using PiB-PET 15 years later [116].

What is more, it was revealed in a prospective study of 165 subjects that the presence of metabolic syndrome accelerated amyloid-beta deposition in patients with pre-identified amyloid-beta deposits on PiB-PET [129].

In the study conducted by the AD Neuroimaging Initiative using 18F-FDG PET, it was observed that AD subjects had reduced glucose uptake in regions associated with AD (posterior cingulate, precuneus, parietal, and frontal cortex) compared to people without cognitive impairment [130].

Even more significantly, glucose metabolism was also assessed using 18F-FDG PET in young PCOS women. The study revealed decreased glucose uptake in the frontal, parietal, and temporal cortex, as well as the hippocampus and amygdala, compared to healthy controls [117]. Additionally, patients with prediabetes and DM2 presented reduced glucose metabolism in regions associated with AD [131]. In a study assessing glucose metabolism using PET with 18F-FDG in healthy subjects, glucose hypometabolism in AD-associated areas correlated positively with glucose and insulin concentrations, independently of IR [118]. The findings suggest that hyperinsulinemia worsens cellular energy efficiency in AD-related areas.

The mechanisms underlying the above observations are supported by the results of experimental studies. Rodents with induced insulin resistance presented increased aggregation of amyloid-beta and tau protein, as well as reduced cognitive ability [132]. One of the causes of amyloid-beta accumulation might be connected to its abnormal clearance, a process that physiologically involves the insulin-degrading enzyme (IDE) [133], which is predominantly found in the endothelial cells of blood vessels within the CNS [134]. Insulin resistance is accompanied by compensatory hyperinsulinemia, which competitively reduces the availability of IDE for amyloid-beta. As a result, amyloid-beta concentration is increased in circulation, which enhances passage to the CNS [135,136]. A study of human tissues from AD patients revealed increased amounts of transcripts for gamma-secretase and decreased amounts of transcripts for IDE. This dysregulation was attributed to increased signaling of the Notch1 pathway, a product of the gamma-secretase metabolic pathway, involved in processes of neuroplasticity, neurogenesis and long-term memory [137]. Additionally, rodent models of AD demonstrated overexpression of amyloid-beta aggregation, CNS inflammation and decreased cholinergic signaling within the hippocampus due to Notch pathway signaling. GLP-1 agonist liraglutide, administered intraperitoneally for 30 days, led to the normalization of Notch signaling and reduced cognitive deficits in subject rodents [138].

A nested case-control study involving a cohort of 176,250 patients from the Danish National Diabetes Register, including 11,619 individuals with dementia, revealed a dose-dependent negative correlation between the use of anti-diabetic drugs (such as metformin, DDP-4 inhibitors, and GLP-1 agonists) and the odds of dementia in diabetic patients [119].

On the other hand, the precise role of insulin in AD pathogenesis remains unclear. Meta-analysis conducted in 2024 showed disturbances in carbohydrate metabolism correlate positively with increased tau protein deposition but not with amyloid-beta deposition [139].

The body of evidence confirms that metabolic disorders, including insulin resistance, are the key pathophysiological links between PCOS and AD.

### 5.2. The Role of Luteinizing Hormone and Anti-Mullerian Hormone (AMH)

Progesterone deficiency observed in PCOS results in increased LH concentration, and LH receptors are found in the central nervous system [140,141]. Additionally, it has been suggested that prenatal exposure to increased levels of maternal androgens influences the abnormal formation of neuroendocrine loops within the hypothalamus, leading to abnormal function of gonadotropin-releasing hormone (GnRH)-producing neurons and consequently increased LH secretion [142]. The Anti-Mullerian hormone (AMH) role in PCOS is still discussed, as the high levels of this hormone are present in the majority of patients [143,144]. Magnetic resonance spectroscopy with tractography was utilized to visualize the increased activity of hypothalamic neurons in PCOS women compared to the general population. Subsequently, the authors of the study demonstrated in an animal model that exposure to AMH affects the increased activity of GnRH-secreting neuroendocrine loops [145]. These findings are consistent with another study, which showed that prenatal exposure to AMH affects the reprogramming of hypothalamic neuroendocrine circuits, contributing to their increased activation and resulting in impaired secretion of GnRH [146].

The function of LH in the AD pathogenesis was also discussed since women suffering from AD presented higher levels of gonadotropins than healthy women. Increased levels of gonadotropins have been observed for decades in women with AD compared to healthy women [147]. What is more, a positive correlation between levels of circulating LH and amyloid-beta was also found in men with AD [148]. The study employing PET-PiB and magnetic resonance imaging showed a positive correlation between LH and FSH levels and cerebral amyloid-beta burden in elderly women [120]. Administration of βHCG, a molecule structurally homologous to LH, in ovariectomized rats treated with estradiol resulted in spatial memory impairments and increased amyloid-beta 1-40 and 1-42 levels [121].

PCOS patients exhibit weaker cognitive performance in various domains [149]. A study assessing brain activity in MRI revealed increased LH correlation with decreased activity in cortical areas responsible for visuospatial memory, while areas responsible for episodic memory showed increased activity [150].

Another study demonstrated that the use of acetylcholinesterase inhibitors in combination with leuprolide acetate, a gonadotropin-releasing hormone (GnRH- analog that reduces LH secretion), for 48 weeks resulted in the preservation of cognitive function in women with moderate to severe AD [151]. Moreover, studies in animal models have shown that administration of leuprolide acetate to transgenic rodents with induced AD resulted in reduced amyloid-beta accumulation and improved cognitive abilities [152]. A study employing neuroimaging also identified altered white matter microstructure in young women with PCOS accompanied by cognitive decline; these changes were independent of BMI and education level [153].

Another commonality between AD and PCOS refers to abnormalities in StAR protein synthesis, which is a rate-limiting enzyme for progesterone (Prog) biosynthesis in mitochondria. In PCOS, decreased production of Prog leads to the abolishment of negative feedback between Prog and LH, resulting in increased LH synthesis. StAR protein transports free cholesterol, a substrate necessary for progesterone production, into the mitochondria [154]. In PCOS, there is a reduction of the StAR protein expression of the StAR protein and the associated reduction in the expression of the CXCL14 chemokine [122]. Similarly, in AD, there is also a decrease in StAR protein expression in the hippocampus, correlating with reduced neuroprotective neurosteroid synthesis and promoting amyloid-beta deposition [123,155]. Moreover, rodent studies have demonstrated an increase in CXCL14 chemokine expression in astrocytes in mice with the BACE1 gene knocked out, resulting in a reduction of amyloid-beta accumulation [156]. The findings of these studies suggest a role of decreased StAR expression and thus increased LH concentration in the development of amyloidopathy.

AD is more prevalent in women who exhibit increased concentrations of gonadotropins due to the abolition of negative feedback with estradiol [157]. Taking into consideration the above-mentioned data and conclusions that LH might promote amyloidopathy, women with PCOS are exposed to this factor much earlier.

On the other hand, increased LH levels correlate with heightened kisspeptin (KISS) neuronal activity and elevated levels of KISS have been observed in PCOS patients [158]. It is postulated that this protein possesses amyloid-beta binding properties, thereby limiting its toxicity in AD [159]. Further research is needed to better explain the cause of the relationships in this complex issue.

### 5.3. Dysfunction of the Kynurenine Pathway

The possible denominator linking metabolic disorders, PCOS, Alzheimer disease, and inflammation is dysregulation of Trp metabolism via the kynurenine pathway. The role of disruption of this pathway in AD has been postulated for many years. Immunohistochemical analysis of tissues taken from the brains of AD patients showed IDO expression and the presence of quinolinic acid (QA) in senile plaques in the hippocampus and intraparietal cortex [160]. Amyloid-beta 1-42 has also been shown to promote QA production by microglia [161]. QA presents in vitro neurotoxic effects by triggering the excitotoxicity process of overstimulation of NMDA receptors, leading to activation of phospholipases, proteases, and endonucleases, which damage cell structure. Moreover, it is proven that QA augments hyperphosphorylation of the tau protein [162].

In the study evaluating levels of tau protein, amyloid-beta, kynurenine, serotonin, Trp, and brain structure via MRI in both AD and healthy subjects, it was proven that an increase in the kynurenine/serotonin ratio (KYN/5-HT) correlated with elevated serum levels of inflammatory markers. Evaluation of tau protein, amyloid-beta, kynurenine, serotonin, and Trp concentrations, and brain MRI images revealed a positive correlation with KYN/5-HT. Although greater deposition of amyloid-beta and tau protein in the brain parenchyma, as well as smaller hippocampal volume, thinner rim, and frontal cortex, and poorer performance on tests assessing cognitive functions were observed in the study group, no relationships between those changes and the KYN/TRP ratio were found [124]. On the other hand, a case-control study revealed elevated levels of QA and KYNA in both plasma and cerebrospinal fluid in the AD patients compared to healthy subjects [125].

Dysregulation of the kynurenine pathway is already observed in the preclinical stage of AD. In a pilot study employing PET using ^18^F-florbetaben, increased plasma levels of kynurenine and anthranilic acid were found in women without established cognitive deficits and with neocortical amyloid-beta load (NAL) [163]. These findings are consistent with another study, in which the presence of NAL was connected to increased concentrations of kynurenine pathway metabolites and amyloid-beta neurofilament light chain in the plasma of patients with preclinical Alzheimer disease [164]. Current evidence strongly suggests a role for Trp metabolism dysregulation in AD, but its exact role remains incompletely elucidated. Despite these doubts, IDO inhibitors are taken into consideration as possible therapy for AD [165,166]. In vitro studies have shown that amyloid-beta stimulates IDO expression in neurons and, as a result, increases the production of neurotoxic kynurenine derivatives [167].

Meta-analysis involving 22 studies with 664 patients diagnosed with AD showed a trend of increased KYNA synthesis in CSF and thus decreased 3-HK concentration in CSF. KYNA, being an NDMA antagonist, exhibits neuroprotective properties; it protects against excitotoxicity mediated by excessive glutamate concentrations. On the other hand, glutamate in physiological concentrations mediates synaptic plasticity, but this process is inhibited at elevated KYNA levels. Intriguingly, there were no revealed differences in QA levels between AD and cognitively healthy patients [168].

Overactivation of the kynurenine pathway also occurs in PCOS. In a Chinese population, increased levels of KYN, KYNA, and neurotoxic QA were described in PCOS women. Shifts in the Trp catabolite profile (TRYCAT) positively correlated with AMH, LH, and fasting glucose levels [126]. In another metabolomic study, the TRYCAT profile was analyzed in the urine of patients with PCOS. Testosterone concentration, LH/FSH ratio, and plasma free androgen index were positively correlated with all TRYCAT in the study group compared to healthy individuals [127]. Additionally, the relationship between Trp level and plasma androgen levels was also described in PCOS [128].

Derangement of the kynurenine pathway might be a contributing factor in the presence of mitochondrial dysfunction found in both PCOS and AD. IDO activity increases as a result of immune cell stimulation mediated by pro-inflammatory cytokines. Therefore, inflammation emerges as a common factor in AD, PCOS, and the overactivation of the kynurenine pathway [169,170]. Taking into consideration that the main product of the kynurenine pathway is NAD(P)H, crucial for proper mitochondrial function, it might be suspected that dysregulation of the kynurenine pathway causes mitochondrial dysfunction [171,172]. Additionally, a positive correlation has been observed between increased IDO mRNA expression and circulating extracellular mitochondrial RNA, which might reflect damage to these organelles [173].

In vitro studies have demonstrated 3-HK increases the production of free radicals that damage mitochondria [174] and leads to aberrant protein synthesis within these organelles, thereby impairing their function. Consequently, an energy deficit appears and, if it becomes a chronic process, may lead to cell apoptosis.

A higher level of 3-HK has been observed in the serum of patients with AD [175] but not in the CSF [168] of patients with AD. No clear explanation of discrepancies between studies related to serum and CSF 3HK level in patients with AD can be found in the literature; however, all research indicates the significant role of the kynurenine pathway in the course of AD. Damage to mitochondrial DNA caused by free radicals leads to aberrant protein synthesis within these organelles, thereby impairing their function. This results in an energy deficit, which, if it becomes a chronic process, can lead to cell apoptosis [176,177]. PCOS is associated with increased oxidative stress, which may contribute to a higher incidence of mitochondrial dysfunction in this group [178]. Additionally, mitochondrial mtDNA mutations are more prevalent in individuals with PCOS, potentially rendering them more susceptible to this pathology. It was established that PCOS patients were significantly more likely to exhibit mt4977 DNA mutations, a marker of mtDNA damage across the entire mitochondrion [179].

Like in PCOS, dysregulation of the kynurenine pathway is also linked to insulin resistance. In an obese population, metabolism within the kynurenine pathway was found to be enhanced in individuals with type 2 diabetes compared to normoglycemic controls, with a positive correlation observed between insulin resistance, assessed with the HOMA index, and 3-HK levels [87]. Moreover, in vitro studies have demonstrated that metformin exerts neuroprotective effects by preventing QA-induced excitotoxicity [180]. Additionally, this drug reduces the phosphorylation of the tau protein [181]. These findings underscore the importance of dysregulation within the kynurenine pathway, mitochondrial dysfunction, chronic inflammation, and oxidative stress in the pathophysiology of both diseases.

### 5.4. Other Biomarkers

A cross-sectional study conducted in 2024 revealed significantly higher plasma levels of amyloid precursor protein (APP), fibronectin (FN), and its fragments FN1.3 and FN1.4, as well as APO-E and von Willebrand factor (vWF) in PCOS patients compared to non-PCOS women. It is noteworthy that the concentration of these proteins is typically elevated in patients with AD [9]. Both AD and PCOS exhibit elevated levels of microRNA-222; however, the significance of this finding remains unclear [182].

Patients with PCOS have been found to have elevated serum leucine levels, which correlate with the degree of insulin resistance [67]. Leucine, an amino acid, stimulates protein synthesis in skeletal muscle. It competes with Trp and kynurenine for transport across the blood-brain barrier (BBB) via the LAT1 transporter; administration of leucine to rodents caused a reduction in the kynurenine/Trp ratio in CSF and mitigated symptoms of inflammation-induced depression [183]. Therefore, it can be hypothesized that the increase in leucine levels in PCOS reflects “leucine resistance”, found also in our analyses [58], resulting in possible disturbances in the transport of particles involved in the kynurenine pathway to CSF.

## 6. Conclusions

Numerous questions regarding the pathomechanisms underlying the two discussed conditions remain unanswered. Current evidence shows that some dysregulation in PCOS, such as metabolic syndrome and insulin resistance, along with excessive activation of LH-secreting neuroendocrine circuits and accompanying chronic inflammation, are positively correlated with amyloid-beta aggregation. These findings suggest that women with PCOS might face a heightened risk of neurodegeneration. It seems that consideration should be given to investigating the epidemiological relationship of the development of AD in patients with previously diagnosed PCOS using archival cohort databases. Moreover, we suggest that the role of kynurenine pathway dysregulation is underestimated in explaining the pathogenesis of both discussed diseases (Figure 3).

Further research from this perspective could enhance our understanding of the processes underlying amyloid-beta and tau protein deposition. Altered proportions of kynurenine pathway metabolites may serve as diagnostic markers for neurodegeneration or for assessing the risk of its occurrence. Ultimately, the development of inhibitors targeting specific enzymes within this pathway might lead to therapeutic interventions.

Both AD and PCOS are the subjects of in-depth scientific research focused on the characteristics specific to these conditions. The aim of this review was to highlight broader pathogenetic phenomena that may aid in understanding the mechanisms underlying the development of PCOS and AD. Investigating the kynurenine pathway appears to offer such possibilities.

## Figures and Tables

**Figure 1 biomolecules-14-00918-f001:**
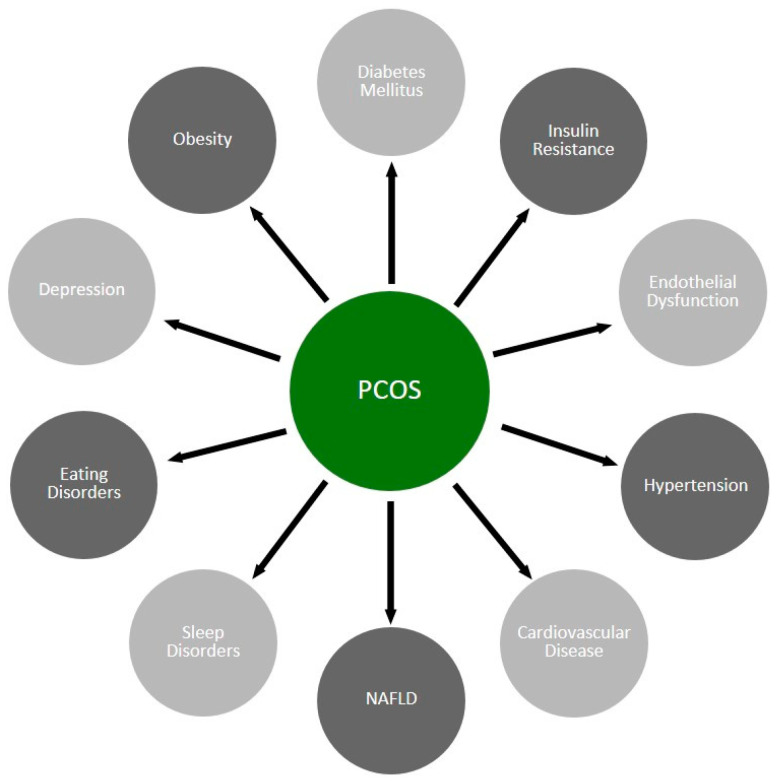
PCOS-related disturbances influencing peripheral and brain metabolism. The presented conditions occur more frequently in patients with PCOS.

**Figure 2 biomolecules-14-00918-f002:**
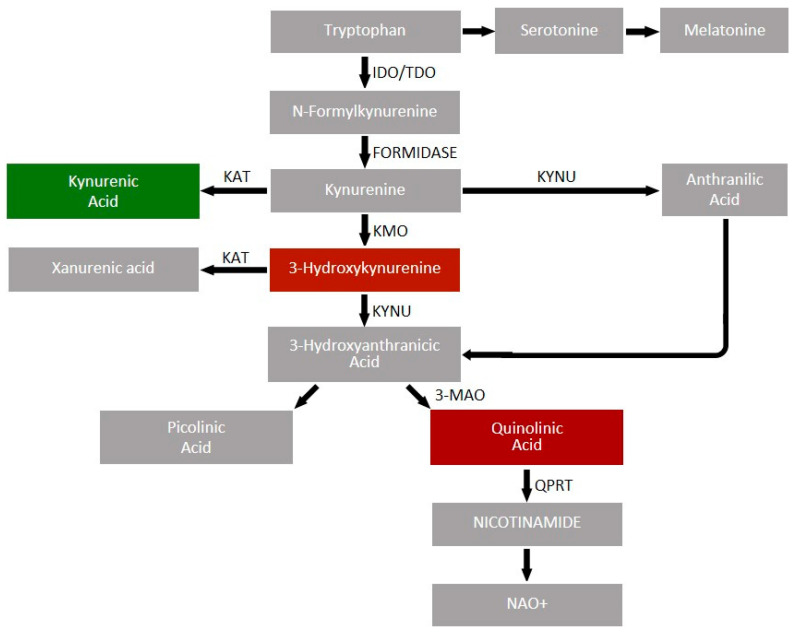
Kynurenine pathway scheme. Different metabolisms in the pathway have diverse impacts on the brain. Red color -metabolite exhibiting neurotoxic properties, green color-metabolite exhibiting neuroprotective properties.

**Figure 3 biomolecules-14-00918-f003:**
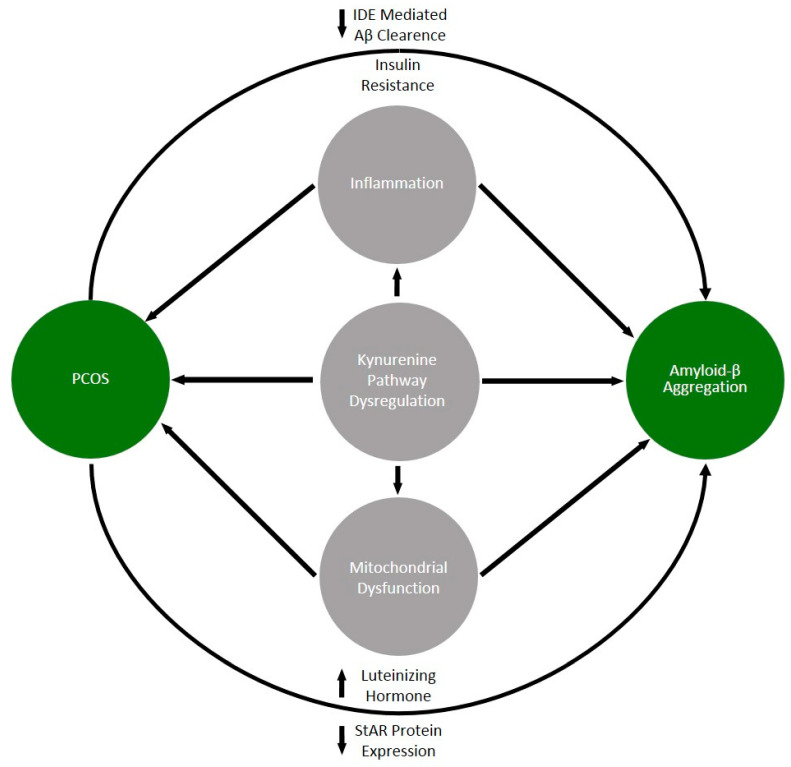
Summary of factors that may contribute to the increased risk of neurodegeneration in PCOS. Arrows indicate the potential direction of the factor’s action.

**Table 1 biomolecules-14-00918-t001:** Summary of findings linking PCOS with AD.

Method	Finding	Model	Putative AD-PCOS Link	Reference
PiB-PET	HOMA positively correlates with Aβ accumulation in AD-related brain areas	AD patients	IR present in PCOS might promote Aβ aggregation	[115,116]
18FDG-PET	Glucose hypometabolism in AD-related brain areas when compared to healthy subjects	PCOS patients	PCOS is a risk factor for cellular energy efficiency in AD-related areas	[117]
18FDG-PET	Glucose hypometabolism in AD-related brain areas positively correlates with insulin and glucose concentration	Healthy subjects	Hyperinsulinemia worsens cellular energy efficiency in AD-related areas.	[118]
Laboratory	Dose-dependent negative correlation between usage of antidiabetic medication and odds of dementia	Diabetic patients	IR and metabolic syndrome increase the risk of dementia	[119]
PiB-PET	LH and FSH positively correlates with amyloid burden	Older-age subjects	Excessive LH secretion present in PCOS might promote Aβ aggregation	[120]
Laboratory	Treatment with bHCG promotes Aβ aggregation	Ovariectomized rats	LH might have amyloidogenic properties	[121]
Laboratory	StAR protein lowered expression in AD	AD patients	Lowered StAR expression leads to increased Aβ aggregation; expression of this protein is lowered also in PCOS	[122,123]
Laboratory	Kynurenine pathway overactivation present in both PCOS and AD	PCOS and AD patients		[124,125,126,127,128]
Laboratory	3-HK levels are positively correlated with HOMA IR	Diabetic patients	Kynurenine dysregulation is linked to IR	[87]
Laboratory	Amyloid precursor protein concentration is increased in PCOS group compared to healthy subjects	PCOS patients	In PCOS Aβ aggregation might be more likely	[11]

## Data Availability

No new data were created or analyzed in this study. Data sharing is not applicable to this article.

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
