# Peer review of "Are Women with Polycystic Ovary Syndrome at Increased Risk of Alzheimer Disease? Lessons from Insulin Resistance, Tryptophan and Gonadotropin Disturbances and Their Link with Amyloid-Beta Aggregation"

_biomolecules, 2024, doi:10.3390/biom14080918_

Round 1

Reviewer 1 Report

Comments and Suggestions for Authors

The manuscript is a narrative review regarding the association of polycistic ovary syndrome and Alzheimer Disease.

It is overall well-written, but I recommend some corrections:

Line 16 to 17: Should be re-written as it is difficult to understand the meaning, maybe due to the repetitions of “...as one of the most common...”

Line 19: Error in ---“certain proteins marker proteins…”

Line 23: Recommend to delete “the issue of”

Line 25: Substitute “the view is presented” by “we concluded”

Line 26: I suggest to say that changes occurring in PCOS may influence the increased risk of neurodegeneration.

Line 55: “and its levels are increased”, instead of “is increased”

Line 56: “its levels rise”, instead of “its level rises”

Line 107: including, but not limited to, mutations (insert commas)

Line 141: Increasing amounts of evidence suggest (not suggests)

Figure 1 – NAFLD is not mentioned in the text before Fig 1 or in the figure as Non-alcoholic fatty liver disease, this information should be added as a legend to the figure, for example.

Line 252: I think PET should be added as an acronym

Line 256: Typo in positively

Line 284: “Enhances” instead of “enhance”

Line 307: “LH receptors” instead of “receptors for LH”

Line 312: “High levels of this hormone are present”, instead of “the increased level of this hormone is present”

Line 350: “of the StAR protein” seems to be repeated

Line 393: These findings are (instead of is) consistent

Line 419: Typo in resultof of

Line 476: I suggest to call the chapter “Conclusion” instead of “Summary”

Comments on the Quality of English Language

The English is good enough to be published, but I've found a few grammatical errors and typos. If they are corrected according to my suggestions, it will be fine.

Author Response

Line 16 to 17: Should be re-written as it is difficult to understand the meaning, maybe due to the repetitions of “...as one of the most common...”

Line 19: Error in ---“certain proteins marker proteins…”

Line 23: Recommend to delete “the issue of”

Line 25: Substitute “the view is presented” by “we concluded”

Line 26: I suggest to say that changes occurring in PCOS may influence the increased risk of neurodegeneration.

Line 55: “and its levels are increased”, instead of “is increased”

Line 56: “its levels rise”, instead of “its level rises”

Line 107: including, but not limited to, mutations (insert commas)

Line 141: Increasing amounts of evidence suggest (not suggests)

Figure 1 – NAFLD is not mentioned in the text before Fig 1 or in the figure as Non-alcoholic fatty liver disease, this information should be added as a legend to the figure, for example.

Line 252: I think PET should be added as an acronym

Line 256: Typo in positively

Line 284: “Enhances” instead of “enhance”

Line 307: “LH receptors” instead of “receptors for LH”

Line 312: “High levels of this hormone are present”, instead of “the increased level of this hormone is present”

Line 350: “of the StAR protein” seems to be repeated

Line 393: These findings are (instead of is) consistent

Line 419: Typo in resultof of

Line 476: I suggest to call the chapter “Conclusion” instead of “Summary”

Reply:  Thank you very much for taking the time to review this manuscript. We really appreciate your opinion about our article.  Thank you for your remarks, we have corrected text in places, where it was pointed out. Thanks to your recommendations, we have done a further review of the article for linguistic mistakes and made additional corrections to the lines 370-371 and 367

Reviewer 2 Report

Comments and Suggestions for Authors

In this review, the authors discussed the issue of the relationship between Alzheimer's disease (AD) and polycystic ovary syndrome (PCOS), particularly focused on the role of disorders of tryptophan metabolism in both conditions. The points of focus are interesting and innovative.

 Comments:

1. In the third section, concerning the overview of PCOS, the authors should add the definition and exploration of its pathogenesis, with particular emphasis on elucidating the mechanisms underlying metabolic abnormalities.

2. In fifth part, I suggest the authors summarize the common denominators of PCOS and AD in table.

3. In the conclusion of Part 6, the current research progress should be summarized, and the prospects and recommendations for research directions related to the association and mechanism of AD risk in PCOS metabolic abnormalities should be proposed.

4. The illustration of each figure should include a brief overview of its content in addition to the title.

Author Response

Thank you very much for taking the time to review this manuscript. We appreciate all given remarks We appreciate all the feedback we have received from you. Please find the detailed responses below.

  1. In the third section, concerning the overview of PCOS, the authors should add the definition and exploration of its pathogenesis, with particular emphasis on elucidating the mechanisms underlying metabolic abnormalities.

Reply: Thank you for this suggestion – we agree that this part should contain more information about PCOS pathogenesis. We have rearranged this section to explain patogenesis of PCOS more clearly, and show factors which might promote occurence of metabolic disturbances in this syndrome. However, extact mechanistical link between disscused disease and metabolic disturbances is still unknow, what we tried to emphasize. In our article we are trying to explore factors which have potential role in explanation of these phenomena.

  1. In fifth part, I suggest the authors summarize the common denominators of PCOS and AD in table.

Reply:  Thank you for this suggestion. Thanks to your comment, the value of the article increases. We created table summarizing most important findings linking AD and PCOS

  1. In the conclusion of Part 6, the current research progress should be summarized, and the prospects and recommendations for research directions related to the association and mechanism of AD risk in PCOS metabolic abnormalities should be proposed.

Reply: Thank you for this recommendation. Although both of this diseases are widely studied, an issue of link between these two conditions is slightly neglected, and current body of research is poor. We have rearranged Part 6 to emphasize what is currently known in this area. We also added our propositions of furhter research.

  1. The illustration of each figure should include a brief overview of its content in addition to the title.

Reply: We have added a brief overview to each figure, in addition to the title.

Round 2

Reviewer 2 Report

Comments and Suggestions for Authors

In this revised manuscript, the authors have made revisions in accordance with the peer review comments. The following are some minor issues that require attention.

1. Table 1, Figure 2, and Figure 3 are not referenced in the original text.

2. Some punctuation formatting issues need to be corrected, such as Line128, Line131, Line228, Line 494.

Author Response

Thank you one more time for taking the time to review this manuscript. We have corrected the issues that were pointed out. Please find the detailed responses below.

  1. Table 1, Figure 2, and Figure 3 are not referenced in the original text. Response: We have referenced Table1 in orginal text in part 5.1. Figure 2 has been referenced in part 4.3, and Figure 3 in part 6.
  2.  Some punctuation formatting issues need to be corrected, such as Line128, Line131, Line228, Line 494. Respone: Thank you for this remark. We have corrected the mentioned mistakes. Thanks to this suggestion, we conducted an additional search for punctuation formatting issues and corrected them.